# Indicators of Land, Water, Energy and Food (LWEF) Nexus Resource Drivers: A Perspective on Environmental Degradation in the Gidabo Watershed, Southern Ethiopia

**DOI:** 10.3390/ijerph18105181

**Published:** 2021-05-13

**Authors:** Zinabu Wolde, Wu Wei, Haile Ketema, Eshetu Yirsaw, Habtamu Temesegn

**Affiliations:** 1College of Land Management, Nanjing Agricultural University, Nanjing 210095, China; sos.zine04@gmail.com (Z.W.); haileketema2005@yahoo.com (H.K.); 2College of Natural Resources, Dilla University, Dilla 419, Ethiopia; eshetu.yirsaw@du.edu.et (E.Y.); habte023@du.edu.et (H.T.); 3Nantional and Local Joint Research Center for Rural Land Reseources Use and Consolidation, Nanjing 210095, China

**Keywords:** land-water-energy-food nexus, driving factor, indicators, degradation, path coefficient

## Abstract

In Ethiopia, land, water, energy and food (LWEF) nexus resources are under pressure due to population growth, urbanization and unplanned consumption. The effect of this pressure has been a widely discussed topic in nexus resource literature. The evidence shows the predominantly negative impact of this; however, the impact of these factors is less explored from a local scale. As a result, securing nexus resources is becoming a serious challenge for the country. This necessitates the identification of the driving factors for the sustainable utilization of scarce LWEF nexus resources. Our study provides a systemic look at the driving factor indicators that induce nexus resource degradation. We use the Analytical Hierarchical Process (AHP) to develop the indicators’ weights, and use a Path Analysis Model (PAM) to quantitatively estimate the effect of the driving factor indicators on the LWEF nexus resources. The results indicate that social (48%), economic (19%), and policy and institutional changes (14%) are the major nexus resource driving factor indicators. The path analysis results indicate that among the social driving factor indicators, population growth and consumption patterns have a significant direct effect on the LWEF nexus, with path coefficients of 0.15 and 0.089, respectively. Similarly, the potential of LWEF nexus resources is also influenced by the institutional and policy change drivers, such as outdated legislation and poor institutional structure, with path coefficients of 0.46 and 0.39, respectively. This implies that population growth and consumption patterns are the leading social drivers, while outdated legislation and poor institutional structures are the institutional and policies change drivers which have a potential impact on LWEF nexus resource degradation. Similarly, other driving factors such as environmental, economic and technological factors also affect nexus resources to varying degrees. The findings of our study show the benefits of managing the identified driving factors for the protection of LWEF nexus resources, which have close links with human health and the environment. In order to alleviate the adverse effects of driving factors, all stakeholders need to show permanent individual and collective commitment. Furthermore, we underline the necessity of applying LWEF nexus approaches to the management of these drivers, and to optimize the environmental and social outcomes.

## 1. Introductions

There has been a growing demand for land, water, energy and food (LWEF) nexus resources in the past half-century due to increasing population growth, urbanization, and unplanned consumption [1,2]. This has pushed more than one billion people to face shortages of land, water, energy and food. The risk of nexus resources insecurity impacts results from the interaction of natural and anthropogenic factors which lead to the vulnerability and exposure of the human and natural system that substantially contributes to nexus resource degradation [3]. This diminishes the capacity of the nexus resources that perform essential functions and services in the ecosystem [4,5].

The concept of the nexus has gained increasing attention in the research and policy making communities [6,7,8]. However, there is an inadequacy in the identification of the driving factors that affect the pool of LWEF nexus resources, which necessitates comprehensive study and critical reflection on the existing nexus Resource Drivers [2,6,9]. The identification of the driving factors of land, water, energy and food is critical to provide useful information for the improvement of the sustainable utilization of nexus resources [1,3,10].

Currently, there is an important need to study the potential negative impacts of LWEF nexus resource drivers from global, national and regional scales [2,9,10,11]. However, the complex relationship existing among these nexus resources makes it difficult to easily understand the synergies between nexus resources and their driving factors. This confounds management and leads to nexus resource degradation [12]. Therefore, understanding nexus resource drivers plays and essential role for sustainable nexus resource management. However, these driving factors vary with the patterns of geography, climate, economic development, social and political integration, and the transformation of the landscape [13,14]. This necessitates site-specific action in terms of management, which can be achieved by identifying the resource driving factor indicators. Those identified indicators are perceived by the public and used as simplified and aggregated forms to present information pertaining to a certain region [15,16,17].

A study on natural resources conservation showed that the demand for productive land, water, energy and food is driven by common driving factors, such as rapid population growth, urbanization and climate change [18]. However, there are many other driving factor indicators affecting nexus resources at different scales. Eventhogh the nexus resource driving factor indicators vary on the basis of small geographical units, the studies conducted so far have been focused at large scale [8]. Hence, there is limited information on the nexus resource driver indicators from the local level perspective, upon which our study intended to focus. In the current study area, there is an indication of LWEF nexus degradation due to different driving factors characterized by multifaceted of land-uses, water, energy access, and food insecurity [18]. Therefore, this study attempts to select nexus resource drivers in relation to the prevailing conditions in the study area, and uses path coefficient analysis to model the causal relationship between the LWEF nexus and drivers. With this, the study intends to: (i) identify LWEF nexus resource driving factor indicators that affect nexus resources, (ii) explore the extent and trends of a driver on four nexus resources, and (iii) identify the direct and indirect impact of driving factors on LWEF nexus resources.

## 2. Material and Methods

### 2.1. Study Area

The Gidabo watershed was chosen as the subject of this study in order to explore the relationship between the land, water, energy and food nexus and its driving factors. It is part of central rift valley of Ethiopia, where agricultural expansion and built-up land is expanding rapidly [19,20]. The expansion of agriculture and build up in this watershed resulted in population growth and increased demand for productive land, water, energy, and food nexus resources.

In central Ethiopia, the main Ethiopian Rift divides the Ethiopian highlands into northeastern and southwestern halves. This rift includes the Ethiopian Rift Valley lakes which occupy the floor of the rift valley, with different sub-watersheds. The Gidabo watershed is located between latitude 6°9′′4″ and 6°56′′4″ N, and longitude 37°55′ and 38°35″ E (Figure 1). It has a peak of 3213 m.a.s.l. and the lowest altitude is about 1171 m.a.s.l.

This watershed is bordered by the catchment of Lake Hawassa to the north, the river Bilate to the west, the river Galana to the south, and Genale–Dawa river basin to the east, which ae considered to be potential livelihood sources for Southern Ethiopia. The populations are settled more towards the eastern highland, and the population density decreases towards the eastern lowlands. There is more population around the eastern highlands, and it reduces as one goes down to the eastern lowlands. However, the population of the watershed has been growing alarmingly in the last three decades, and it is currently more than 1.5 million. The upper part is more populated (>500 inhabitants per square kilometer), and this has an immense impact on potential nexus resources. The livelihoods of the local community in the study area primarily depend on mixed farming and livestock rearing [21], which are sensitive to land, water, energy and food availability. Therefore, due to the current nexus resource degradation by various driving factors, most farmers in the watershed live with the lowest access to electricity, clean water and modern infrastructure, and they are vulnerable to frequent food insecurity.

### 2.2. Data Sources and Techniques

This study focuses on the driving factor indicators of the LWEF nexus, which were identified from varied literature sources and a deep survey into the history of the nexus resources’ statuses. Additionally, data were collected from key informants and the local community using a combination of structured interviews. Secondary data sources were also used for the validation of the LWEF nexus indicators. The identification of the driving factor indicators helps us to have a common understanding and to easily categorize which LWEF nexus resources are directly or indirectly affected by the drivers. The survey was conducted within a four-month period from July 2019 to October 2019, following two approaches. First, expert interviews and focus group discussions (*n* = 50) were conducted with respondents from the natural resource, agriculture, water and energy sectors in order to characterize the driving factors (Table 1). Second, questionnaires were distributed to a total of 434 households in order to find out how the local community perceives the identified driving factor indicators. To perform this, the respondents were asked to rank each predefined indicator’s impact on the LWEF nexus from 0 to 4 (4 = extreme, 3 = high, 2 = medium, 1 = low, 0 = negligible).

### 2.3. Selection of the LWEF Nexus Resource Drivers

According to [14,22], driving factors are defined as driving forces consisting of different components that can affect societal change or natural systems. Understanding driving factors helps us to identify the status, development, and management of LWEF nexus resources in order to ensure equity and sustainability [23]. The driving factor indicators are variables that describes the status of the nexus resources and their impact on the availability and distribution trends of nexus resources. In nexus resource management, a single indicator cannot efficiently describe a complex process that affects land, water, energy, and food degradation [22]. Therefore, the indicators are combined to create a composite index to monitor the state of the nexus resources. Environmental indicators may be considered as a simplified way to present information on a certain region [24]. Therefore, in our current study, we consider nexus resource drivers’ indicators in order to clearly understand area-specific driving factors (Table 1).

The nexus resource driving factor variables were selected concerning the objectives and the prevailing conditions in the case study area using relevant literature and expert opinions. Therefore, based on the objective of the study, LWEF nexus resources and driving factor indicators were identified with the following stepwise process: (1) Collecting ideas, which can be performed by compiling all of the ideas from key informants without judging them, then organizing the group members to categorize them based on specific individual objectives and analytical questions. (2) Structure and refine the ideas: in this step, we further structure and consolidate the ideas to sort out the relevant ones by referring to the work of other researchers or using a previously developed set of indicators. During this step, some unnecessary indicators could be rejected, and others with similarities could be merged. (3) The formulation of the indicators, to make sure that the selected indicator shows what results are to be reached within which target group, and in what time frame. (4) The selection of the indicators: here, we assembled too many indicators in steps 1 through 3. Because the indicators’ quality is more important than their number, we set priorities to have a small but meaningful set of indicators. Following this, AHP and PCM were used to develop the weight value for each individual driving factor indicator (Appendix A Table A1). This weight value could be analyzed by the path coefficient analysis model in order to evaluate the direct and indirect effect of the driving factors on LWEF nexus resources. Finally, the researchers and experts from the four sectors were organized into a group, in order to provide feedback on the interim results of the factors affecting the nexus resources. Table 1 provides the five main and twenty-five sub-driving factor indicators and their explanations.

During the identification and grouping of the indicators, particularly the sub-indicators, we tried to reduce the overlap and mix-up of the sub-indicators in order to reduce the multicollinearity of the sub-indicators. During our field investigation, the local community chose 47 different driving factor indicators. Following this, a discussion was held among experts from the agriculture, water and energy, natural resource, and environmental management sectors to score and rank the sub-indicators, and to categorize them into the main indicators. After careful evaluation, the experts identified 25 sub-driving factor indicators.

To verify this result, we computed the multicollinearity for all of the identified indicators using Variance Inflation Factor (VIF) analysis. The finding of the analysis revealed that 22 indicators had strong correlation with each other; therefore, they were removed from the analysis to reduce multicollinearity. Additionally, the computed mean Variance Inflation Factor (VIF) was 3.4, which is satisfactory to show that there is no multicollinearity. Therefore, 25 sub-indicators were finally identified and categorized into five main driving factor indicators. Besides this, we also defined each sub-indicator in order to reduce the data-based multicollinearity, which can be removed using subject-area knowledge and factors in the goal, as reported by [7,23].

**Table 1 ijerph-18-05181-t001:** LWEF nexus resource driving factor indicators in the study area.

S/No	Main Driver Indicators	Sub-Driver Indicators	Code	Description	Literature
1	Social	Population growth	SC1	A growing population will increase the use of natural resources.	[3,19,25,26,27,28,29]
		Poverty	SC2	Poverty is increasingly recognized as an important driver of forest which affect WEF system.	
		Lack of alternative livelihoods	SC3	Lack of alternative livelihoods leads to little stake in the health and productivity of natural resources.	
		Consumption patterns	SC4	Consumption patterns fairly convincingly explains the dynamics of poor approach to natural resources and their resource use behavior.	
		Community awareness	SC5	Ignorance of local community knowledge is becoming both limitation of their environmental resource and consequence of their using practices.	
		Lack of Public involvement	SC6	Evolving technical and institutional measure to prevent over-extractive resource use.	
2	Economic	Increasing income variability	EC1	Natural resources provide important services to both local on-site and off-site beneficiary, while most off-site beneficiary are “free rider”, this related with income variation.	[2,30,31,32,33,34]
		Low capital	EC2	Lack of allocation of sufficient capital investment for resource rehabilitation and control leads to degradation, Because capital budget provides an important tool for the control and evaluation of resources.	
		Increasing WEF prices	EC3	Implication of raising energy prices linked supply of firewood and charcoal, this induces pressure on land resources.	
		Increasing land value	EC4	In recent decades, alarming land value leads to strong land speculation and grabbing, in which expansion of small, large and unplanned industries affect nexus resources.	
		Inadequate financial resources	EC5	Low funding level to restore degraded LWEF nexus resources both from government and NGO’s results on unwise and open use of nexus resources.	
3	Institutional and policy change	Outdated legislation	IP1	There were overall agreement in policy formulation and documentation, however updating and implementing with the pace of resource degradation have inconsistences.	[10,34,35,36,37,38,39,40,41]
		Inadequate financial capital	IP2	Inadequate financial capital is characterized by high quality LWEF nexus institution which leads to higher rate of innovation and interlinked business formulation.	
		Poor institutional structure	IP3	LWEF resources can potentially contribute to development outcomes, but nowadays those resources are plagued with unsustainability, poor governance, corruption and conflict of interest which lead to degradation.	
		Poor stakeholder network	IP4	Stakeholder analysis can be used to avoid inflaming conflicts among land, water, energy and food sectors, and ensure that the marginalization of certain groups is not reinforced, and fairly represent diverse interest.	
4	Environmental	Fuel wood dependence	EN1	As populations is increasing from time to time, there would be a massive wood fuel shortage and that an increasingly desperate population would move into untouched forests, causing massive deforestation.	[9,35,41,42,43,44,45,46]
		Charcoal production	EN2	Charcoal production has greater environmental cost. It is made by burning large logs in kilns or in mounds of earth to create low-oxygen environment, this leads land degradation which affect water, energy and food.	
		Agricultural expansion	EN3	Agricultural developments are an important driving force behind developments and the organization of society as a whole, which often results in intensive dynamic land-use changes.	
		Land use change	EN4	Land use change encompasses different types of land use expansion in the expense of LWEF.	
		Climate change	EN5	Climate change creates critical challenges with increasing temperature, agro-ecological change, and changing precipitation for water, energy, and food, as well as ecosystem processes.	
		Industrial expansion	EN6	Industrial expansion poses serious challenges in the use of land, water and other NRs.	
5	Technology	Lack of input supply	TC1	Technological input supply increase productivity in agriculture, efficient water and land use.	[46,47,48,49,50,51,52,53,54,55]
		Inadequate technology adoption and implementation	TC2	In developing nations, millions lack access to sanitation services and safe drinking water, modern energy sources and optimized land use.	
		Attitude towards technology innovation and development	TC3	Lack of proactive attitudes towards technology efficiency, adoption and implementation results on unwise resource use.	
		Lack of infrastructure	TC4	There is growing momentum to address traditional and emerging threats to the LWEF resources through innovative technology infrastructure.	

### 2.4. Analytical Hierarchical Process and Pairwise Comparison Matrix

The data for the LWEF nexus resource indicators were collected from the experts using the Analytic Hierarchy Process (AHP), which helps to normalize the indicators and establish the indicators’ weights. The Pairwise Comparison Matrix (PCM) was used to determine the weighted ranking of the indicators [56].

The PCM is constructed by using scores that represent the experts’ judgment in order to compare and measure the importance of the indicators in relation to all of the other indicators using the Saaty scale [57], in which the relationships are established using a scale ranging from 1 to 9, and their reciprocals are established using the question “How important is two identified indicators to drive nexus resources?” Appendix A Table A1 shows the results of the pairwise comparison matrix of the LWEF nexus indicators, as computed using the following formula:(1)$$C_{v}=\overset{n}{\underset{i=1}{\sum }}X_{i}K_{ir}$$
where $C_{v}$ indicates the composite value of the indicators, $K_{ir}$ is the relative change index of the driving factor indicators, $\mathrm{and}\;K_{i}$ and $X_{i}$ are the weight of the indicators. Then, through further calculation weighting, the criteria and indicators within each criterion were computed, as outlined by [10]. Based on the weight value of the driving factor indicators, the highest weight was computed for social, economic, institutional and policy change, while the lowest weight was computed for environmental and technological driver indicators (Figure 2).

Calculating the composite weight of the driving factor indicators provided a starting point of analysis, and was used as a summary indicator to guide policymakers and other data users [58]. From the social driving factors, population growth (SC1), poverty (SC2), and lack of public involvement (SC6), and from institutional and policy change, inadequate financial capital (IP2) and poor institutional structure(IP3) indicators have a strong impact on nexus resource degradation (Appendix A Table A1).

### 2.5. Path Coefficient Analysis Models

Models have an important role in organizing data and information. They also help to guide the identification of the indicators that meet the required goal [59]. Specifically, an indicator-based model provides an overview for the consideration of environmental problems. In this study, the path analysis model (PAM) is used to quantitatively analyze the direct and indirect impact of the LWEF nexus resources drivers.

A path coefficient can be expressed as a standardized coefficient, which is typically calculated by standardizing all of the variables and then computing the path coefficients from the ratios of the standard deviation of the variable using Stata 14. Standardized coefficients allow direct comparisons of the magnitude of the effects of two causal variables measured on different scales [57]. A path model may look similar to a multiple regression [60], where an exogenous variable is analogous to a predictor variable, and an endogenous variable is a response; however, the difference is that the endogenous variable can be both a predictor and response in a system of equation. An important assumption is that exogenous variables are measured without error, On the other hand, each endogenous variable is assumed to have error.

In path analysis, the correlation co-efficient which defines a standardized coefficient is partitioned into direct and indirect effects of independent variable (i.e., driving factors) on the dependent variable (i.e., LWEF nexus resources). In order to estimate the direct and indirect effects of the correlated variables, we took $k_{1\;}$, $k_{2}$ and $k_{3}$ as the driving factors and *n* as its effect on the nexus resources; a set of simultaneous equations is required to be formulated, as shown below:(2)$$Y_{nK_{1}}=P_{nK_{1\;\;}}+P_{nK_{2\;}}Y_{k1k2}+P_{nK_{3}}\;Y_{k1k3}$$
(3)$$Y_{nK_{2}}=\;P_{nK_{1\;}}Y_{k1k2\;}+P_{nK_{2\;\;}}+P_{nK_{3}}\;Y_{k2k3}$$
(4)$$Y_{nK_{2}}=\;P_{nK_{1\;}}Y_{k1k3\;}+P_{nK_{3}}\;Y_{k2k3}+P_{nK_{3\;\;}}$$
where, P denotes the path coefficients and Y denotes the simple correlation co-efficient. The total correlation between $k_{1}$ and *n* is thus partitioned as follows: $P_{nk_{1}\;}$ = the direct effect of $k_{1}$ on *n*, $P_{nk_{2}Y_{k1k2\;}\;}\;$ = the indirect effects of $k_{1}$ via $k_{2}$ on n, $P_{nk_{3}y_{k1k2}}$ = the indirect effects of $k_{1}$ via $k_{3}$. Finally, the Stata14 statistical package was used for the analysis.

## 3. Result and Discussion

### 3.1. Analysis of the Land, Water, Energy and Food (LWEF) Nexus Driver Factor Indicators

This section introduces the main and sub-driving factors that affect one or more of the land, water, energy, and food nexus systems. We identified twenty-five sub-driving factor indicators, which were categorized under the five main driving factor indicators. The results of the analysis indicated that from the five main driving factor indicators, social (48%), economic (19%), and institutional and policy changes (14%) are the major driving factors that affect nexus resources (Figure 3). Similarly, the average weight of the main nexus resource driving factor also shows the same trends (Figure 2). The direct and indirect impacts of these driving factors are explained in the subsequent section.

Social life changes, low economic sources, and outdated institutional and policy changes are major driving factors of land, water, energy, and food nexus security. Nowadays, the limited understanding of those driving factor results in poor LWEF nexus resource management [6].

#### 3.1.1. Social Drivers

Social drivers relate to the social structure and institutions that shape people’s preferences, behavior and possibilities, and the capacity of individuals and groups to influence the environmental system. In our study, we identified six social driving factor indicators (Table 1).

Figure 4 indicates the path coefficient analysis results of the social driving factor indicators that affect nexus resources: population growth (SC1), poverty (SC2), the lack of alternative livelihoods (SC3), consumption patterns (SC4), community awareness (SC5) and the lack of public involvement (SC6). The results of the path co-efficient analysis revealed that population growth (*p =* 0.15), lack of alternative livelihoods (*p =* 0.56) and lack of public involvement (*p =* 0.15) had a positive direct effect on the LWEF nexus resources (Figure 4). Population growth also affects the LWEF nexus indirectly (*p =* 0.005), which is mediated by consumption patterns, while poverty had a negative indirect effect (*p =* −0.16) on the LWEF nexus, which was mediated by consumption patterns (Figure 4).

The results of the path analysis revealed that community awareness (SC5) had a positive direct effect (*p =* 0.096) on LWEF nexus degradation in the study area (Figure 4). This implies that a lack of community awareness in nexus resource management disrupts the planning and execution of nexus resource management. Community awareness, which is linked with a bottom-up approach, is important for the success and failure of resource management. Community awareness towards natural resources ensures greater control over the ecological and socio-cultural aspects of sustainability, as well as a broader and more responsible analysis of the livelihood benefits for resource users and local residents.

The demographic development and characteristics in the study area have implications for the high demand on the existing patterns of land, water, energy and food availability, and strongly driven nexus resource potential. The rural and urban population in the study area increased from 2005 to 2020, and is also expected to increase up to 2035 with increasing demand for nexus resources (Figure 5). This population growth may also increase the levels of poverty, which results in the unwise use of land, water, energy and food.

Rapid population growth can undermine natural resource conservation at the national level, and is associated at the local level with lower status and opportunity for the rural poor. The relationship between population growth and nexus resource demand is changing and becoming complex due to increased consumption and the use of nexus resources both in urban and rural areas (Figure 5). Similarly, a larger population means higher consumption, which in the long run puts increased pressure on nexus resources. It also shows, in the coming decades, that rapid urbanization will accelerate land use changes which load pressure on WEF nexus resources.

In Ethiopia, serious social and environmental challenges of urbanization remain unsolved in many urban areas. These challenges can be exacerbated by natural resource degradation and rapid urban growth in regions and cities. Figure 5 shows that the demand for LWEF nexus resources mount in the coming decades, and this will affect the sustainability of nexus resources.

#### 3.1.2. Economic Drivers

Economic development, in the past, has been a driver of increased resource use and environmental damage. According to [61], the household consumption of goods and services over a life cycle accounts for about 60% of the total environmental impact from consumption. An increase in nexus resource scarcity is linked with economic well-being. Low economic potential lowers the quality and quantity of nexus resources, which causes human beings to degrade nexus resources [62]. Figure 6 depicts five economic nexus resource driving factor indicators.

The weight of the driving factor indicators shows that increasing income variability (EC1) and increasing WEF prices (EC3) are the dominant economic driving factors that affect nexus resources in the study area (Appendix A Table A1). The results of the path coefficient analysis indicate that increasing income variability (*p =* 0.044) and increasing WEF prices (*p =* 0.057) have positive direct impacts on LWEF nexus resources (Figure 6).

Regarding low capital (EC2), it has both direct (*p =* 0.12) and negative indirect (*p* = −0.003) impacts on LWEF nexus resources (Figure 6). This necessitates that governments set capital to stimulate economic activity to meet particular sectorial development goals. For low capital, the characteristics of low economic potential have taken the brunt of the blame by causing the unsustainable use of nexus resources, which is caused by low investment. These economic activities have greater risk for people who are dependent on the natural resource sector; agricultural workers, pastoral and forest communities, and those experiencing multiple forms of inequality, marginalization and poverty are most exposed to the impacts.

Therefore, the identification of the pathway by which the economic drivers impact LWEF nexus resource degradation requires an understanding of direct economic drivers, and their solution through prioritization. In order to incorporate this understanding, the pathway between LWEF nexus degradation and economic driving factor indicators needs to be mapped (Figure 6).

Increasing land value (EC4) indirectly drives (*p =* 0.009) LWEF nexus resources, which are mediated by increasing income variability (Figure 6). The improper land use practices affect the availability of water, energy, and food, which then have environmental effects. This implies that low economic development affects the economic capacity to invest in degraded nexus resource restoration. With this, the economic benefits of nexus resources for the local community have recently changed dramatically, and have led to the overexploitation and unsustainable utilization of nexus resources.

#### 3.1.3. Environmental Drivers

A critical challenge for the environment over the coming decades is the demand for food production with scarce land, water, and energy sources [63]. The decline of those nexus resources will be riskier for food production in the future than in the past due to complex driving factors.

Environmental driving factors have a complex character which experiences different impacts on LWEF nexus resources. As the pressure in the environment grows, the nexus resources might change; consequently, there might be an increase in the degradation process. According to [26], there may be different environmental driving factor indicators. For example, the climate is a component of environmental driving factor indicators; it has become an independent driver of environmental change and poses a serious challenge to future natural resource management. Table 1 shows the environmental driving factor indicators that affects nexus resources. These are fuel wood dependencies (EN1), charcoal production (EN2), agricultural expansion (EN3), land use change (EN4), climate change (EN5), and industrial expansion (EN6).

The results of the path coefficient analysis indicate that industrial expansion (*p* = 0.326) exhibits the highest magnitude of direct effects on the LWEF nexus, followed by climate change (*p* = 0.194) and agricultural land expansion (*p* = 0.172) (Table 2). These driving factors are considered to be the principal environmental driving factors that affect the LWEF nexus. In Table 2, fuel wood dependence has a negative direct effect (*p =* −0.038) and a positive indirect effect on the LWEF nexus, which is mediated by charcoal production (*p* = 0.291), agricultural expansion (*p =* 0.059), land use change (*p =* 0.12), climate change (*p =* 0.066) and industrial expansion (*p =* 0.065).

As depicted in Table 2, the use of charcoal as an energy source has a negative direct (*p* = −0.038) impact on LWEF nexus resource degradation, but it affects it indirectly via agricultural expansion (*p* = 0.001), land use change (*p =* 0.018), climate change (*p =* 0.075), and industrial expansion (*p =* 0.018). This indicates that the use of charcoal as an energy source may affect nexus resources based on the extent of consumption; however, its integration with other factors may result in unintended impacts. For example, land cover change induced by massive charcoal production affects land, water, energy and food production potential [7]. According to [22], fuel wood collection and charcoal production is the main land resource driver in sub-Sahran Africa, and the same is true in the current study area.

#### 3.1.4. Technology as a Driver

Technological developments necessitate higher levels of productivity in terms of the use of land, water, and energy to increase food production. Technology can be expressed through various indicators; however, the current study identified four indicators based on the experts’ judgment and literature, such as the lack of input supply (TC1), inadequate technology adoption and implementation (TC2), the attitude towards technology innovation and development (TC3), and the lack of infrastructure (TC4).

Based on the weight of the indicators, the lack of input supply (TC1) is ranked as the most important technology driver that causes LWEF nexus resource degradation in the study area (Appendix A Table A1). The lack of access to technological input supply results in the unproductive use of land, water, and energy resources [64]. However, technological processes may also produce unwanted results, such as the loss of biodiversity, ecosystem disturbance and increased deforestation, which create a trade-off in the LWEF nexus. Table 3 shows that TC1, TC2 and TC3 have a significant correlation with LWEF nexus resources, while TC4 does not, which implies that TC4 does not have a significant direct impact on the LWEF nexus. Similarly, the path coefficient analysis shows that inadequate technology adoption and implementation (*p =* 0.198) shows the highest magnitude of a direct impact on LWEF nexus resources, followed by the attitude towards technology innovation and development (*p =* 0.079) (Table 3). This implies that inadequate technology adoption and implementation, and negative attitudes towards technology from the local community can limit the proper use of land, water, and energy resources. Nowadays in Ethiopia, due to lack of site-specific technology extension land wastage, irrigation water loss and energy disruption are becoming serious problems, in agreement with study of [65].

Table 3 highlights that the attitude towards technology innovation and development has a significant indirect impact on nexus resources, which is mediated by TC1 (*p =* 0.001), TC2 (*p =* 0.818) and TC4 (*p =* 0.012). This implies that people are causing environmental changes—notably in the biosphere, hydrosphere, and atmosphere—which are associated with a lack of technology-based input supply, inadequate technology, and a lack of infrastructure. These changes are the result of human activities linked with time and space, leading to global environmental problems.

In general, technological advances have created unintended consequences that make it difficult to determine whether the advances have long-term positive and/or negative impacts. This also plays a critical role as an instrument for the observation and monitoring of the environment on global and local scales, in agreement with the findings of [66].

#### 3.1.5. Institutional and Policy Change as a Driver

The alignment of institutional structures and policies are vitally important to utilize nexus resources effectively, efficiently, and equitably. In Ethiopia, the institutional structure and policy to integrate and manage the LWEF system as one component is at an infant stage, and there is a compliment of one sector to the other. This creates complex problems of relevance, quality, accessibility and equity [64], which result in land, water, energy, and food insecurity.

Even though there are institutions and policies working on Ethiopian forests, in the last three decades, the forest cover of the country dropped from 40% to 3% [67]. This is linked with unintended institutional and policy changes which induce land degradation and affect WEF nexus security across the country.

The causes of the failure in nexus resource security were linked with frequent institutional and policy changes. These changes became LWEF resource drivers (Table 1). Based on the literature review and consultation with the experts, we consider outdated legislation (IP1), inadequate financial capital (IP2), poor institutional structure (IP3), and poor stakeholder networks (IP4) as the common institutional and policy change indicators that affect the LWEF nexus.

According to [10,68], the nexus approach is becoming gradually more prominent on policymakers’ agendas at the global and national level. However, in Ethiopia particularly, in the case study area, inclusive outdated legislation on the land, water, energy and food sectors is challenging. This could be due to those sectors applying different concepts and contrasting types of interaction from the budget allocation, management scope, and organogram. Such differences prevent shared understanding, which manipulates LWEF resources.

The result of the analysis indicates that outdated legislation (*p* = 0.24) and poor institutional structures (*p* = 0.15) directly affect LWEF nexus resources (Figure 7). The lack of updating regulations on the basis of time and resource use, with the current era of increasing population, disrupts the balance between resources and resource usage. Inadequate financial capital indirectly affects LWEF nexus resources through outdated legislation (*p =* 0.089) and poor institutional structure (*p* = 0.008). These results show that most programs, projects and activities did not aim to set in place the conditions under which local communities will be economically willing and able to conserve LWEF resources, resulting in nexus resource degradation by placing unsustainable demands on natural resources.

Poor stakeholder networks affect nexus resources indirectly through outdated legislation (*p =* −0.026) and poor institutional structures (*p =* 0.07), as highlighted in Figure 7. In the last two decades, many policies and projects designed in the current study area have not achieved LWEF nexus security due to poor stakeholder networks. The lack of understanding of institutional and policy changes as driving factors influences nexus resource consumption, regulation and management, which results in negative externalities across sectors.

Figure 8 reveals comprehensive model fitness statistics for the five main LWEF nexus resource driving factor indicators. The results of the analysis indicate that except for institutional and policy change, other main driving factors directly affect LWEF nexus resources. Additionally, economic, social, and environmental driving factors also indirectly affect LWEF nexus resources.

The result of the path coefficient analysis revealed that economic (*p =* 0.025) and social (*p =* 0.011) driving factors had a positive direct effect on LWEF nexus resource degradation (Figure 8). This implies that LWEF resource degradation often results from immediate causes, such as economic crisis and unsustainable resource management practices, or due to underlying causes including population density, poverty, a lack of alternative livelihoods, and consumption patterns.

Institutional and policy change indirectly affect LWEF nexus resources by enhancing or reducing technological advancement. Similarly, social structures and characteristics need a binding policy in order to regulate unsustainable resource utilization activities that contribute to natural degradation.

### 3.2. Impact of LWEF Nexus Degradation on the Socio-Economy, Livelihoods and Ecology

Nexus resource degradation refers to the impairment of the natural quality and quantity of nexus resources which affect human well-being [69]. This is a common problem in Ethiopia, particularly in the case study area, which is characterized by declining land productivity, water availability and energy sources, and continuing food insecurity.

In the study area, there are losses of biological and economic productivity caused by the complexity of land uses, water, energy access, and food insecurity. This could be associated with rapid population increase, urbanization and climate change, which will pose huge pressure on the socio-economy, ecology and livelihood of the local community [70].

#### 3.2.1. Socioeconomic Impact

Land, water, energy and food resources are critical for the development and survival of societies [71]. However, the access to these resources is complicated by the factors driving nexus resource degradation. According to Mohamed [54], following the complexity of LWEF nexus resources, more than 18.1 million people in Ethiopia are forced into frequent nexus resource insecurity, which will affect different socioeconomic components.

The result shows that the access to/availability of food (34.6%), overall electric supply (26%), and poor health conditions (10.57%) are key socioeconomic characteristics that are strongly affected by LWEF nexus degradation in the study area (Table 3). LWEF nexus degradation affects the access to/availability of food by 34.6% compared with other socioeconomic characteristics (Table 4). This implies that land use change, and water and energy crisis challenge food security under changing climatic conditions. Understanding the sustainable management of land, water and energy is one of the keystones for establishing livelihood security, and maintaining the provision of ecosystem services and adaptive capacity against nexus resource degradation.

Socioeconomic characteristics such as age (8.11%), poor health conditions (10.57%), and level of education (8.85%) are affected by LWEF nexus resource degradation (Table 4). Nexus resource degradation affects the socioeconomic conditions of human wellbeing, characterized by poor health conditions, low levels of education and low food supply [19]. The current study area has good water potential compared with other nexus resources; however, the irrigation water potential was reduced by 7.1% due to land use change (Table 4). This causes the population in the study area to continuously depend on food aid, supported by USAID [72].

#### 3.2.2. Ecological Impact

Due to the nexus resource degradation in the study area, the ecological balance was disturbed, which was characterized by the loss of biodiversity and the migration of rural people to search for additional land, water and food.

Land, water, energy and food degradation also interrupts the regulating and provisioning services of the ecosystem, in particular through agro-ecological variations, climate change, forest loss, gully formation, soil erosion, the loss of biodiversity, and the drying of wetlands [73]. During our field investigation, the farming community reported that land use change, water potential reduction, the lack of fuel wood, and declining food production was observed in the last four decades. Following this ecological variation in 2017, frost occurrence was observed in the highland part of the study area that affected the major livelihoods of the local community by lowering the productivity of crops, particularly coffee and *Enset* (Figure 9).

Climate change caused by nexus resource degradation reduces the ecological efficiency of the supply of productive land, sufficient water, safe energy and food (Figure 10). The study area has been subjected to drastic ecological change attributed to climate change, forest loss, the loss of biodiversity and low agricultural productivity, which affect the overall environment (Figure 10).

Threats to ecology arise because of changes in the quality and quantity of the LWEF nexus which underpins all ecological processes [19], and the lack of stakeholders to invest in nexus resources results in the low conservation of biological diversity and ecological integrity.

Figure 10 reveals that climate change and low water potential, forest loss and soil erosion, gully formation, the loss of biodiversity, and low agricultural productivity are all characteristics of ecological degradation, which have potential impacts on LWEF nexus resources [74]. Climate change is one of the basic drivers which results in forest loss, soil erosion, gully formation and the loss of biodiversity. This strongly affects land use, water, energy, and food nexus resources in the study area, and also implies that different ecological characteristics could result in the degradation of LWEF nexus resources and the overall ecosystem [70].

#### 3.2.3. Livelihood Impact

Table 5 implies the livelihoods of rural communities entirely depend on the availability and consumption of land, water, energy and food nexus resources, which directly or indirectly affect livelihood activities [75,76]. We used a structured household survey to explore the ways in which livelihoods are affected by nexus resource degradation. Table 5 summarizes the response of 434 households on how LWEF nexus degradation affects livelihoods.

As shown in Table 5, crop production (38.1%), livestock rearing (16.3%), and agroforestry practices (10.4%) were the major livelihood activities affected by LWEF nexus resource degradation in the study area. We were unable to obtain exact details about the size and extent of the impacts during the interview. However, crop production and livestock are potential livelihoods are affected by nexus resource degradation. Spiegelberg, Baltazar [30] reported that the sustainable management of the WEF nexus is one of the keystones for establishing livelihood security.

Beekeeping (5.29%), fishing (9.9%) and fruit production (7.5%) are also important livelihoods activities affected by the degradation of nexus resources (Table 5). Therefore, LWEF nexus resource degradation hampered the livelihoods of rural communities, and is becoming a major challenge for sustainable development.

Most rural poor are smallholders practicing low-input agricultural production, which needs a substantial amount of productive land, water and energy. The lack of these nexus resources could have triggered a reduction in food production, energy, and water access [75].

Energy insecurity is a significant challenge in the study area, which affects the expansion of schools, health centers and institutional structures, among other things. Furthermore, the lack of available land, water and energy affects small enterprises by 12.5% (Table 5), which is a source of income for the rural community [77].

Due to a combination of driving factors and human influences, rural livelihoods are characterized by extreme uncertainty and the seasonality of land, water, energy and food nexus resources, as reported by [19].

Generally, policymakers in the land, water, energy and food sectors need to gauge the influence of the driving factors on LWEF nexus resources in order to identify the problems and carry out effective strategies. In this context, the identification of the drivers of the LWEF nexus is essential not only for small geographical unity, but also for the sustainable management of basic nexus resource potential in the long run.

## 4. Conclusions

Nowadays, the accelerated global socio-environmental changes to the land, water, energy and food nexus have received increasing attention among academia and policymakers aiming to restore nexus resources. However, the identification of nexus driving factors was limited, and can result in the degradation of nexus resources.

In this study, we identified land, water, energy and food nexus resource driving factor indicators to show the concept of the current nexus resources trade-off. The analysis of the weight of the nexus resource indicators shows that social (population growth, poverty, lack of alternative livelihoods, etc.), economic (increasing income variability and WEF prices), and institutional and policy change (outdated legislation) are the major drivers of nexus resource degradation. This indicates that a lack of understanding of the social, economic, and frequent institutional and policy changes induces LWEF nexus resource degradation.

We argue that many nexus discourses focus on population growth and urbanization as a major nexus resource driver. However, these drivers were studied from a macro scale and with regard to growing megacities for the last few decades, but the variability of these drivers from region to region was under-studied, and needs policy attention. Therefore, this study focused on understanding the social, economic, environmental, institutional and policy change, and technological drivers which induce degradation.

The results of the analysis indicate that, in the study area, LWEF nexus resources are significantly influenced by eight direct driving factors (such as SC1, SC2, EC1, EC2, EC4, EN3, EN4, and TC2), while the other affect it indirectly. These analyzed driving factors and their effects could help stakeholders to better plan and design relevant policies to keep the synergy and trade-off for sustainable nexus resource management.

The findings of this study can also share experiences for nexus resource driving factors from a local to a national scale, which can be of interest to the audience of nexus resource managers and planners to provide a better opportunity to reduce the pressure on LWEF nexus resources. The management of the identified driving factors is essential to protect LWEF nexus resources, which have a close link with human health and the environment. Addressing the adverse effects of driving factors needs all stakeholders to make a permanent individual and collective commitment to protect the environment and reduce the impact of the driving factors on the environment and public health. Furthermore, we underline the necessity of applying LWEF nexus approaches to manage those drivers, in order to optimize the environmental and social outcomes. For future research, different driving factor indicators and their impacts can be used to define benchmarks to identify the requirement of a sustainable LWEF nexus for different parts of the country.

## Figures and Tables

**Figure 1 ijerph-18-05181-f001:**
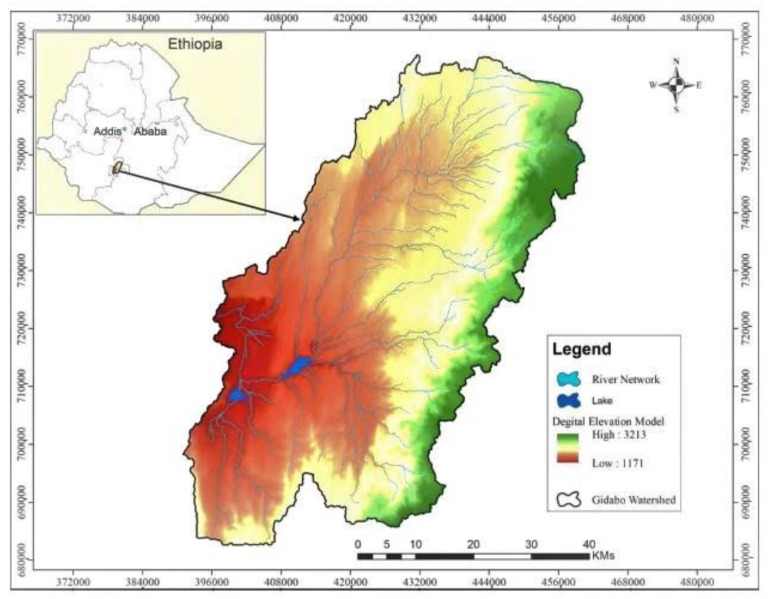
Map of the case study area.

**Figure 2 ijerph-18-05181-f002:**
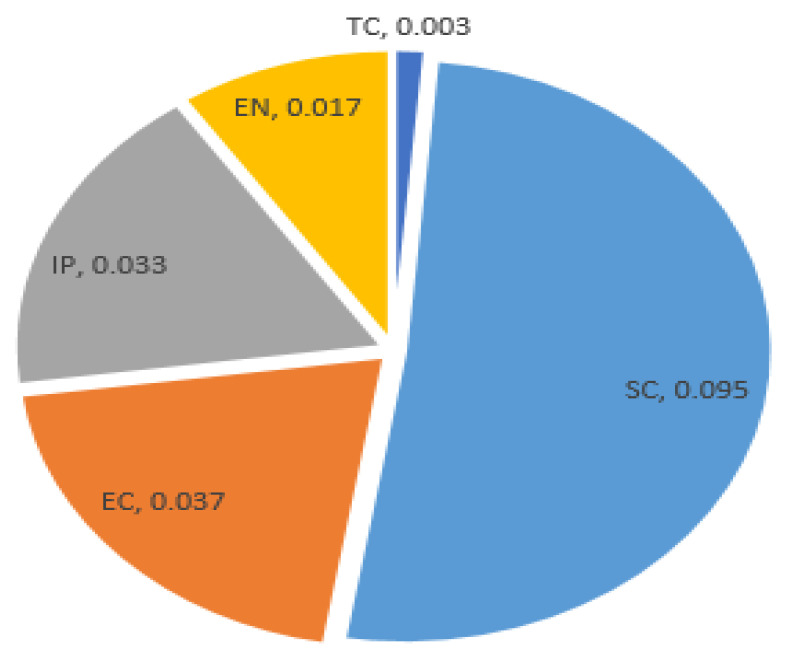
The weight of the five main nexus resource driving factor indicators.

**Figure 3 ijerph-18-05181-f003:**
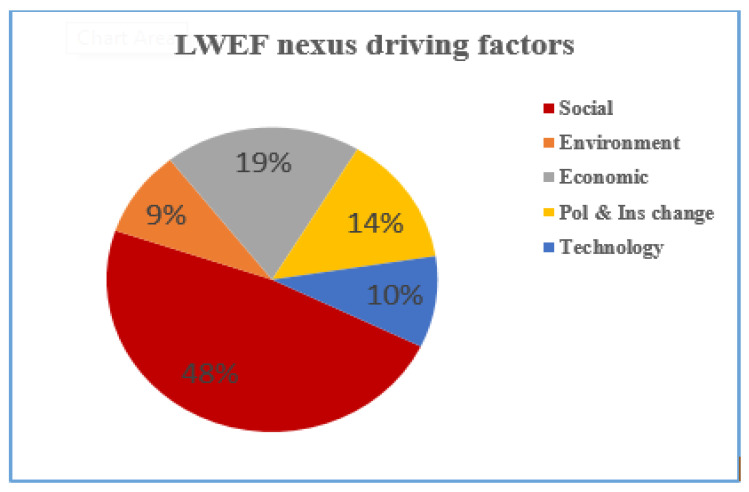
Main LWEF nexus resource driving factor indicators.

**Figure 4 ijerph-18-05181-f004:**
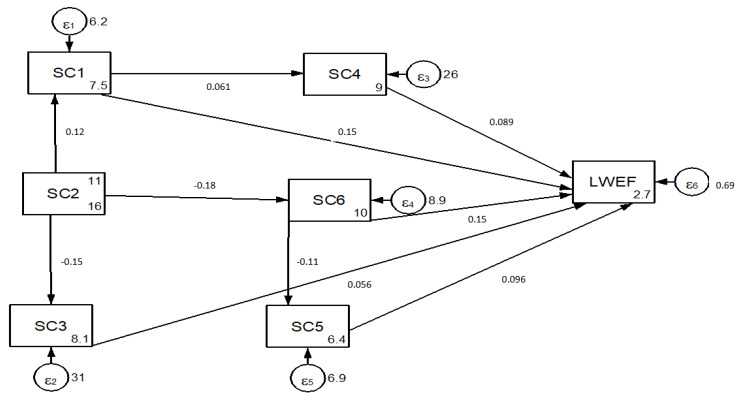
The social driving factor indicators of LWEF nexus resources.

**Figure 5 ijerph-18-05181-f005:**
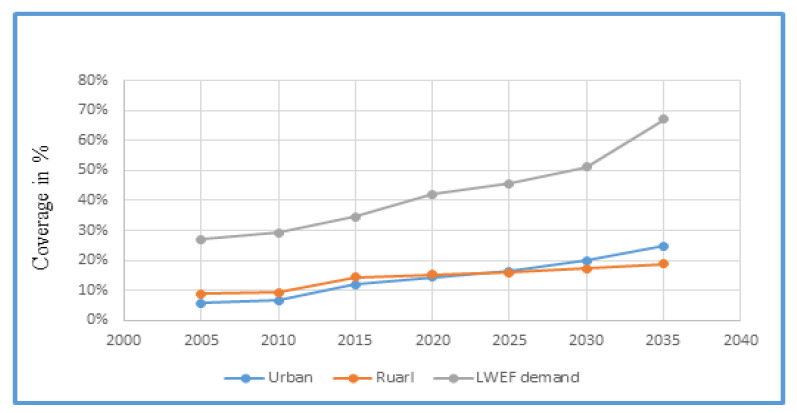
Population projection in the study area (data source: Central Rift Valley document).

**Figure 6 ijerph-18-05181-f006:**
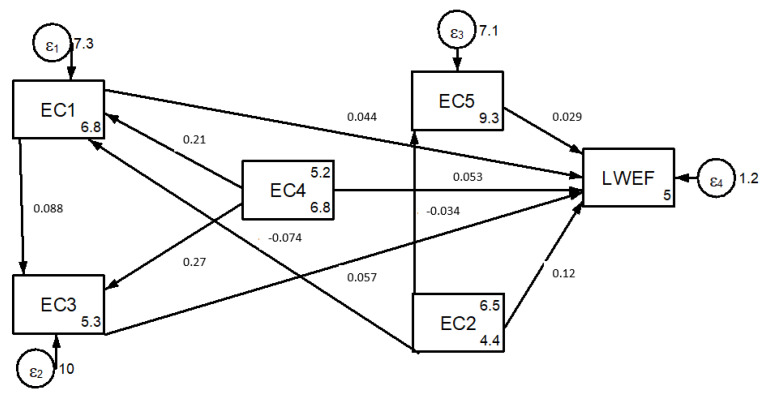
Economic driver indicators of LWEF nexus resources.

**Figure 7 ijerph-18-05181-f007:**
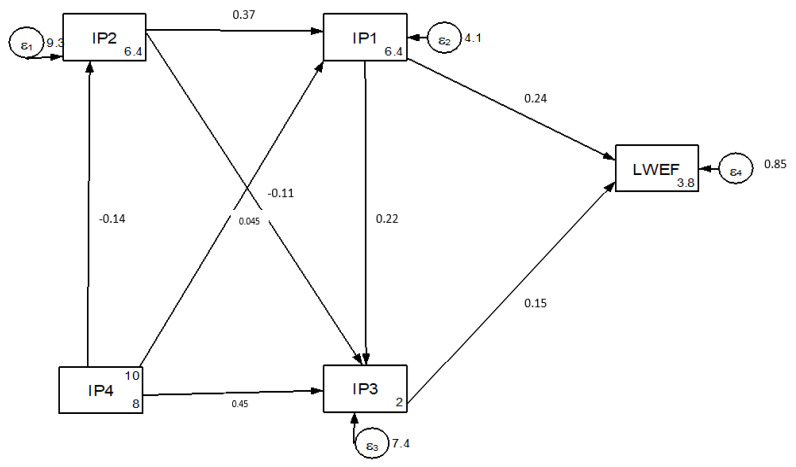
Institution and policy change drivers of LWEF nexus resources.

**Figure 8 ijerph-18-05181-f008:**
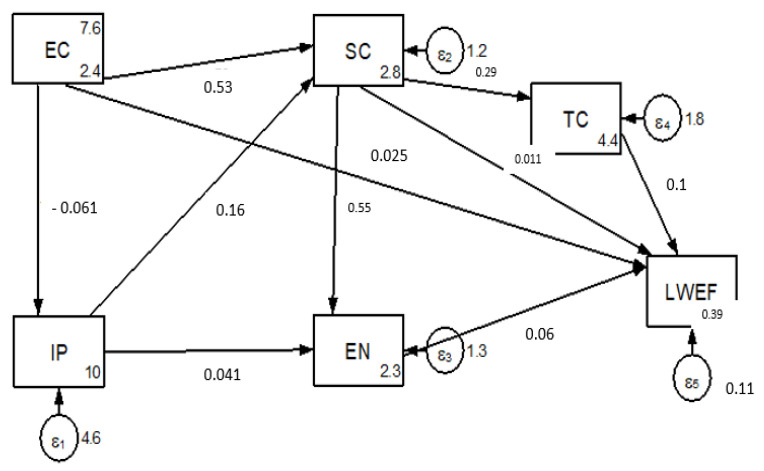
The final comprehensive path model which tested the ways in which the main driving factor indicators affect the LWEF nexus. The indices of model fit demonstrated its excellent goodness-of-fit ($X_{\left(6\right)}^{2}$ = 7.24, *p* = 0.31, TLI = 0.99; NFI = 0.92; RMSEA = 0.03), and all of the paths in the model are significant at *p* < 0.05.

**Figure 9 ijerph-18-05181-f009:**
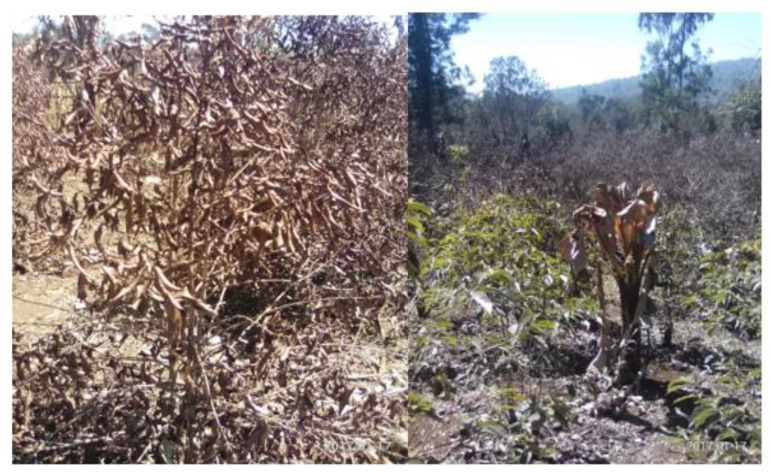
The impact of frost due to climatic variability in the highland part of the Gidabo watershed.

**Figure 10 ijerph-18-05181-f010:**
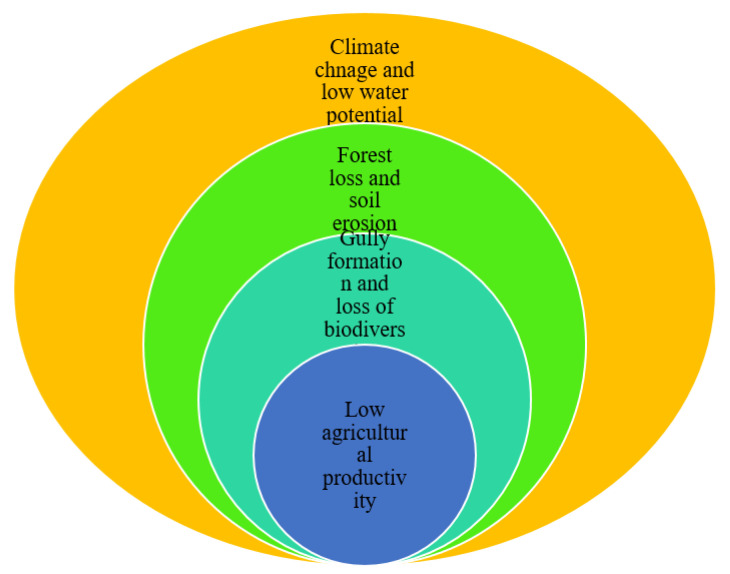
LWEF nexus resource degradation impacts on ecological characteristics.

**Table 2 ijerph-18-05181-t002:** Path coefficient analysis showing the direct (bold) and indirect effects of six causal environmental driving factor indicators on LWEF nexus resources.

Indicators	EN1	EN2	EN3	EN4	EN5	EN6	r	R^2^
EN1	−0.038	0.291	0.059	0.12	0.066	0.065	0.712 *	0.809 *
EN2	0.06	−0.075	0.001	0.018	0.075	0.018	0.669	
EN3	0.012	−0.304	0.172	−0.021	−0.009	0.089	0.876*	
EN4	0.052	0.054	−0.060	0.147 *	0.004	0.120	0.342**	
EN5	0.290	0.038	0.021	0.089	0.194 **	−0.04	0.571	
EN6	0.216	−0.053	0.048	0.073	0.006	0.326 **	0.432	

** Values are significant at *p* ≤ 0.01; * values are significant at *p* ≤ 0.05. EN1 = fuel wood dependence, EN2 = charcoal production, EN3 = agricultural land expansion, EN4 = land use change, EN5 = climate change, EN6 = industrial expansion.

**Table 3 ijerph-18-05181-t003:** Path coefficient analysis for the direct (bold) and indirect effect of technology indicators as LWEF nexus resource drivers.

Indicators	TC1	TC2	TC3	TC4	r	R^2^
TC1	−0.299 **	0.001	0.015 **	−0.031	0.907 **	0.907 *
TC2	0.014	0.198 **	0.002	−0.002	0.147 **	
TC3	0.001 *	0.818 *	0.079 *	0.012	0.316 **	
TC4	0.04	−0.012	0.001	0.023	0.215	

** Values are significant at *p* ≤ 0.01; * values are significant at *p* ≤ 0.05. TC1 = lack of input supply, TC2 = inadequate technology adoption and implementation, TC3 = attitude towards technology innovation and development, TC4 = lack of infrastructure.

**Table 4 ijerph-18-05181-t004:** Impact of LWEF nexus degradation on the socioeconomic characteristics in the study area.

S/No	Socio-Economic Characteristics	Mean	Standard Dev.	% of Impact
1	Age	9.2	2.67	8.11
2	Gender	7.4	2.76	0.14
3	Population density	6.53	2.1	0.04
4	Economic capacity of household	7.39	3.31	0.04
5	Level of education	5.6	2.61	8.85
6	Poor health condition	10.28	2.83	10.57
7	Access to productive land	8	3.07	0.04
8	Overall electric supply	4.97	3.07	26.0
9	Access to clean water	8.65	2.31	4.5
10	Institutional development	8.42	3.06	0.004
11	Access/availability of food	8.78	2.53	34.6
12	Access to irrigation water	10.6	3.94	7.1

**Table 5 ijerph-18-05181-t005:** An overview of the livelihoods activities affected by LWEF nexus degradations which were assessed using household surveys (N = 434).

Livelihoods Activities	Mean	Standard Deviation	Frequency (%)
Crop production	23.75	2.97	38.10
Agroforestry	10.8	2.4	10.40
Livestock	18.95	4.98	16.30
Beekeeping	11.29	6.98	5.29
Fishing	13.21	6.18	9.90
Fruit production	9.05	4.78	7.50
Small-enterprise	12.24	4.03	12.50

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
