# Peer review of "Indicators of Land, Water, Energy and Food (LWEF) Nexus Resource Drivers: A Perspective on Environmental Degradation in the Gidabo Watershed, Southern Ethiopia"

_ijerph, 2021, doi:10.3390/ijerph18105181_

Round 1
Reviewer 1 Report
The paper presents an interesting study about nexus resources drivers, with an environmental degradation perspective in Gidabo Watershed, Southern Ethiopia. It is a suitable topic for International Journal of Environmental Research and Public Health MDPI journal. I recommend a revision following my comments below.
GENERAL COMMENTS:
- The link between the methodology and the results may be improved;
- The figures have low resolution.
SPECIFIC COMMENTS:
- Lines 157-160: This sentence is confusing. It may be revised;
- Line 182: no indentation is needed;
- Line 222: no indentation is needed;
- Line 385: Please correct “Table 3.”;
- Figure 7 was firstly mentioned after its appearance;
- Figure 10 is missing;
- Lines 536-540: These sentences should be revised.
Author Response
Dear Reviwer;
We would like to thank the for you kind review of the manuscript. You raise an important issues and your comments are very helpful for improving the manuscript. We agree with almost all their comments and we have revised our manuscript accordingly.

Reviewer 2 Report
Thanks to the authors for addressing my previous comments. The revised version is much improved and the authors have made substantial revisions.
Author Response

(The authors gave the same response as above.)

Reviewer 3 Report
The manuscript has been improved very much and may be considered for its onward publication
Author Response

(The authors gave the same response as above.)

Round 2
Reviewer 1 Report
The paper presents an interesting study about nexus resources drivers, with an environmental degradation perspective in Gidabo Watershed, Southern Ethiopia. It is a suitable topic for International Journal of Environmental Research and Public Health MDPI journal. I recommend a revision following my comments below.
- Lines 146-161: The added sentences need to be grammarly revised;
- Line 196: no indentation is needed;
- Line 235: no indentation is needed;
- Figures 4, 6, 7 and 8 still have low resolution;
- Figure 7 was firstly mentioned after its appearance;
- Lines 548-549: The sentence should be revised.
Author Response
Dear Reviewer,
Thank you very much for your constructive comments, considering your comment we try to improve our manuscript better than before.

This manuscript is a resubmission of an earlier submission. The following is a list of the peer review reports and author responses from that submission.
Round 1
Reviewer 1 Report
The paper presents an interesting study about nexus resources drivers, with an environmental degradation perspective. It is a suitable topic for International Journal of Environmental Research and Public Health MDPI journal. I recommend a revision following my comments below.
GENERAL COMMENTS:
- The Abstract should be carefully revised and better structured. The main goals of this paper and contributions to the literature should be highlighted. It is mostly focusing on results, which should also be mentioned, but better linked with the goals and future contributions.
- The link between the methodology and the results might be improved.
- The sections are not numbered in a correct order.
SPECIFIC COMMENTS:
- Figures 1, 2, 3, 5, 6, 7, and 9 have low resolution;
- Table 1 should be better explained. What does each indicator mean?
- There is no mention to Figure 2 in the text;
- Figure 5 is placed before figure 4;
- Figure 5, 6 and 9 should be more detailed in the text;
- Line 421: There is an empty “()”.
Author Response
Dear Reveiwer,
Thank you for your comment. We consider your comment as very important to revise our mansucritp. I have highlighted the comment that include in the revised document. Due to the system I only upload your feedback, if you need the revised document I will resend it.
tahnks,
Stay safe.

Reviewer 2 Report
I suspect that multicollinearity may be present among the indicators especially the sub-driver indicators. Multicollinearity should be examined (and addressed if present) before carrying out the analyses.
It unclear how the percentages were derived in lines 185-187 including Figure 3. They add up to 100% so does that mean the driving factor indicators identified in the study are all and the only indicators affecting LWEF?
Why path diagrams are provided for SC, EC, and IP but not EN and TC? Similarly, why path coefficient analysis tables are provided for EN and TC but not SC, EC, and IP?
The path diagrams in Figure 6 and Figure 7, where are (or how to read) the indirect effects shown? What do the numbers mean in the indicator boxes? Also, it’s confusing to read the coefficients in Table 2 and Table 3 regarding direct and indirect effects without a path diagram. For example, what is the indirect effect of EN1 on LWEF? Direct and indirect effect (as well as the total effect) of each driver indictor should be summarized and reported in those tables.
Having separate path analysis done for each main driver indicator is fine but I thought there should be a comprehensive path model that include all the main driver indicators and LWEF. That would allow the authors to analyze how main driver indicators interplay with one another as well as with the LWEF.
The following are editing-related notes:
Line 36 why “WEFL” here but “LWEF” elsewhere?
Lines 123-128 duplicate lines 110-115.
Line 134 shows section 1.4 and then section 2.6 in line 158. Review and redo the section numbers.
Author Response
Dear Reveiwer,
Thank you for your comment. We consider your comment as very important to revise our mansucritp. I have highlighted the comment that include in the revised document. Due to the system I only upload your feedback, if you need the revised document I will resend it.
thanks,
Stay safe

Reviewer 3 Report
The title may be modified as “Indicators of land, water, energy and food (LWEF) nexus resources drivers: A perspective of environmental degradation in Gidabo Watershed, Southern Ethiopia”
Abstract
The authors are highly suggested to have a thorough read out from a native English speaker or a Professional Editing service. There are various typos and grammatical mistakes. Below are the few examples:
Line 17 “environmental indicator” my be modified as “environmental indicators”
Line 21 “The result of the analysis indicates” my be modified as “The results of the analysis indicate”
Line 23 “The path analysis result reveals”
Line 24 “positive significant direct effect”
Line 26 “such as Outdated legislation”
Line 27-28 “This implies population growth, consumption pattern, policies and institutions change drivers LWEF nexus resources which lead to degradation” couldn’t under. Please make it sensible
Line 28-29 “While other driving factors like, environmental, economic and technology also drives nexus resources and varies” have various grammatical mistakes
What is the conclusion?
After reading this sentence, “the identified nexus resource driver in this paper needs further investigation in order to better protect and restore degraded nexus resources.” I would highly suggest the authors to submit this manuscript when they have found something
Author Response
Dear Reveiwer,
Thank you for your comment. We consider your comment as very important to revise our mansucritp. I have highlighted the comment that include in the revised document. Due to the system I only upload your feedback, if you need the revised document I will resend it.
thanks,
Stay safe
Response for each comments
- Reviewer comment-#1. The reviewer asks the title to be modified as “Indicators of land, water, energy and food (LWEF) nexus resources drivers: A perspective of environmental degradation in Gidabo Watershed, Southern Ethiopia
Response to comment: Dear editor thank you for your comment, but we are not clear how we can revise the title, if you may need to revise title by incorporating the quotation, we will reconsider it.
- Reviewer comment-#2. In the abstract the reviewer highly suggested to have a thorough read out from a native English speaker or a Professional Editing service. There are various typos and grammatical mistakes. Below are the few examples:
Response to comment-#2. Dear reviewer thank you for your comments, sorry for same grammatical error, so we consider all your comment accordingly. We revised the whole abstract in our revised document.
Abstract. In Ethiopia, natural resources are increasingly under pressure, especially due to population growth, urbanization, economic growth and other relevant factor. As a result, securing land, water, energy, and food (LWEF) nexus resource becoming serious challenges. This necessitates identification and management of driving factors for sustainable utilization of scarce LWEF nexus resources. What drives those resources however, still needs further research. Our study provides a systemic look at the driving factor indicator that induce nexus resource degradation. We use Pairwise Comparison Matrix and Path Analysis Model to quantitatively estimates the effect of driving factor indicators on LWEF nexus resources. The result of the analysis indicates that, social (48%), economic (19%) and policy and institutional change (14%) are the major nexus resource driver indicators. The path analysis result reveals that from social driving factor indicators population growth, and consumption patterns have positive significant direct effect with the path coefficient of 0.15 and 0.089, respectively. Similarly, the potential of LWEF nexus resources is also influenced by the institutional and policy change driver such as Outdated legislation and poor institutional structure with the path coefficient of 0.46 and 0.39, respectively. This implies population growth and consumption pattern are the leading social driver, while outdated legislation and poor institutional structure are the leading policies and institutions change drivers of LWEF nexus resources. Similarly, other driving factors indicator like, environmental, economic and technology also drives nexus resources with varying amount. Our finding contributes how managing identified driving factor it is essential to protect LWEF nexus resources, which have close link with human health and the environment. Addressing the adverse effects of driving factors need all stakeholder to make a permanent individual and collective commitment to protect the environment and reduce impact of driving factor. Furthermore, we underline the necessity of applying LWEF nexus approaches to manage those driver, in order to optimize environmental and social outcomes.
Specific comments
- Reviewer comment-#2. The reviewer asks to correct Line 17 “environmental indicator” my be modified as “environmental indicators”
Response to comment: Dear reviewer thank you for your comment, we modified the comment as per your request (see our new document page 1, line 14-19)
- Reviewer comment-#3. The reviewer asks to correct Line 21 “The result of the analysis indicates” my be modified as “The results of the analysis indicate”
Response to comment: Dear reviewer thank you for your comment, we modified the comment as per your request (see our new document page 1, line 21)
- Reviewer comment-#3. The reviewer asks to correct the phrase found Line 23 “The path analysis result reveals”
Response to comment: Dear reviewer thank you for your comment, we modified the comment as per your request (see our new document page 1, line 23)
- Reviewer comment-#4. The reviewer asks to correct the phrase found Line 24 “positive significant direct effect”
Response to comment: Dear reviewer thank you for your comment, we modified the comment as per your request (see our new document page 1, line 24)
- Reviewer comment-#5. The reviewer asks to correct the phrase found in Line 26 “such as Outdated legislation”
Response to comment: Dear reviewer thank you for your comment, we modified the comment as per your request (see our new document page 1, line 26)
- Reviewer comment-#6. The reviewer asks to correct the sentence found Line 27-28 “This implies population growth, consumption pattern, policies and institutions change drivers LWEF nexus resources which lead to degradation” couldn’t under. Please make it sensible
Response to comment: Dear reviewer thank you for your comment, we modified the comment as per your request (see our new document page 1, line 27-29)
- Reviewer comment-#7. The reviewer asks to correct the grammatical error found in Line 28-29 “While other driving factors like, environmental, economic and technology also drives nexus resources and varies”
Response to comment: Dear reviewer thank you for your comment, we modified the comment as per your request (see our new document page 1, line 30)
- Reviewer comment-#8. The reviewer asks to revise the conclusion part
Response to comment: Dear reviewer thank you for your comment, we modified the conclusion part by adding the main concern of study and its future aspect (see our new document page 1, line 31-37).
“................Similarly, other driving factors indicator like, environmental, economic and technology also drives nexus resources with varying amount. Our finding contributes how managing identified driving factor it is essential to protect LWEF nexus resources, which have close link with human health and the environment. Addressing the adverse effects of driving factors need all stakeholder to make a permanent individual and collective commitment to protect the environment and reduce impact of driving factor. Furthermore, we underline the necessity of applying LWEF nexus approaches to manage those driver, in order to optimize environmental and social outcomes”.
Thank you for all your comment which helps to better structure our manuscript.
Stay safe

Round 2
Reviewer 1 Report
The paper presents an interesting study about nexus resources drivers, with an environmental degradation perspective. It is a suitable topic for International Journal of Environmental Research and Public Health MDPI journal. I recommend a revision following my comments below.
GENERAL COMMENTS:
- The Abstract has been largely modified. Nevertheless, it is still necessary to perform an overall revision of it. The authors respected my request to highlight the main goals of this paper and contributions to the literature, however a careful revision of the sentences and their links would still be required.
- The link between the methodology and the results might be improved.
SPECIFIC COMMENTS:
- Figures 4, 6, 7, 8 and 10 still have low resolution. It is not only a matter of dpi (if the original figure is not of good quality, it will not solve the problem to just take that figure and improve dpi);
- Lines 140-150 of this new version should be grammarly revised;
- There is still no mention to Figure 2 in the text. Wouldn't "Figure 3" be replaced by "Figure 2" at line 166 of this new version?
- Figure 4 (which was previously numbered as Figure 5) should be more detailed in the text;
- Lines 230-239 of this new version should be grammarly revised;
- Line 360: which "figure"?
- Lines 359-369 of this new version should be grammarly revised;
- Lines 432-438: The sentences should be revised;
- Line 475: There is an empty “()”. It is the same observation that I made for line 421 of the previous version.
Reviewer 2 Report
ijerph-1012133-peer-review-v2
While I appreciate the authors taking the time to respond to my earlier comments, I still find the results and presentation of their path analyses inadequate and confusing. I will use the authors’ response to further elaborate my question and comment.
Response to comment-#3. Dear reviewer thank you for your comment, the reason why me perform path diagram for SC, EC and IP but not for EN and TC are intentional, because the result explained in path diagram for SC, EC and IP and the result explained in Table 2&3 provide the same concept. In the path diagram the direction of the path arrow shows the direct and indirect impact of driving factor on LWEF nexus. Similarly, the bold and diagonal number in table 2&3 shows the direct effect of driving factor indicator on LWEF nexus, while the other numerical vale explains the indirect impacts.
1. The authors’ intention to do path diagrams and coefficients for some selected drivers but not others is not justified. In the path diagram, I can see the direct and indirect effects but my concern is how the authors calculated the effects (especially the indirect effects). In specific, what and where are the indirect effects (in numeric form) in the SC, EC, IP path diagrams? Is it very difficult to include a path coefficient analysis table for SC, EC, IP? Without it, I am not able to verify and/or connect how the reported coefficients in the path diagram are used to derived the numeric values of direct and indirect effects. 2. The path diagrams are very difficult to read because there are a lot of reported statistics. I’ll use Figure 6 as an example, what is the value “6.8” in the EC1 box? Why there are two numbers in the EC4 box “5.2” and “6.8”, what are they, and what do they mean? Why there are two numbers in the EC4 and EC2 boxes but there is only one number in the other boxes? What are the numbers and their meaning next to the epsilon circles?
Reviewer 3 Report
After looking at the revised manuscript there are still many comments which have not been incorporated e.g. title is the same as it was earlier and many others
The authors include various nexus factors but use the word indicator instead of indicators. The authors have used both words and there is non-uniformity in its usage
Moreover, I just read the abstract which was not understandable and has many typos and grammatical mistakes. What about the rest of the portion of the manuscript?
Please see line 475
Also, the results have already been discussed, Is there any need to add a separate portion of the discussion?
There are still many typos and grammatical, that's why it was suggested to be read by a native English speaker or professional editing service
Overall, not satisfactory